# Zinc and Silicon Nano-Fertilizers Influence Ionomic and Metabolite Profiles in Maize to Overcome Salt Stress

**DOI:** 10.3390/plants13091224

**Published:** 2024-04-28

**Authors:** Abbas Shoukat, Zulfiqar Ahmad Saqib, Javaid Akhtar, Zubair Aslam, Britta Pitann, Md. Sazzad Hossain, Karl Hermann Mühling

**Affiliations:** 1Institute of Soil and Environmental Sciences, University of Agriculture, Faisalabad 38040, Pakistan; abbas.shoukat@uaf.edu.pk (A.S.); sarcuaf@gmail.com (J.A.); 2Institute of Plant Nutrition and Soil Science, Kiel University, Hermann-Rodewald-Str. 2, 24118 Kiel, Germany; bpitann@plantnutrition.uni-kiel.de (B.P.); sazzadmh.aha@sau.ac.bd (M.S.H.); 3Department of Agronomy, University of Agriculture, Faisalabad 38040, Pakistan; zauaf@hotmail.com; 4Department of Agronomy and Haor Agriculture, Faculty of Agriculture, Sylhet Agricultural University, Sylhet 3100, Bangladesh

**Keywords:** salinity, physiology, ionomic, metabolites, nano, symplast, apoplast, molecular, subcellular level

## Abstract

Salinity stress is a major factor affecting the nutritional and metabolic profiles of crops, thus hindering optimal yield and productivity. Recent advances in nanotechnology propose an avenue for the use of nano-fertilizers as a potential solution for better nutrient management and stress mitigation. This study aimed to evaluate the benefits of conventional and nano-fertilizers (nano-Zn/nano-Si) on maize and subcellular level changes in its ionomic and metabolic profiles under salt stress conditions. Zinc and silicon were applied both in conventional and nano-fertilizer-using farms under stress (100 mM NaCl) and normal conditions. Different ions, sugars, and organic acids (OAs) were determined using ion chromatography and inductively coupled plasma mass spectroscopy (ICP-MS). The results revealed significant improvements in different ions, sugars, OAs, and other metabolic profiles of maize. Nanoparticles boosted sugar metabolism, as evidenced by increased glucose, fructose, and sucrose concentrations, and improved nutrient uptake, indicated by higher nitrate, sulfate, and phosphate levels. Particularly, nano-fertilizers effectively limited Na accumulation under saline conditions and enhanced maize’s salt stress tolerance. Furthermore, nano-treatments optimized the potassium-to-sodium ratio, a critical factor in maintaining ionic homeostasis under stress conditions. With the growing threat of salinity stress on global food security, these findings highlight the urgent need for further development and implementation of effective solutions like the application of nano-fertilizers in mitigating the negative impact of salinity on plant growth and productivity. However, this controlled environment limits the direct applicability to field conditions and needs future research, particularly long-term field trials, to confirm such results of nano-fertilizers against salinity stress and their economic viability towards sustainable agriculture.

## 1. Introduction

In recent decades, soil salinization in agricultural lands has emerged as a critical issue, adversely impacting global food security [1]. Salinity stress, primarily resulting from natural processes and anthropogenic activities such as irrigation with saline water and land misuse, poses a significant threat to crop productivity [2]. The excessive accumulation of salts in the soil not only hampers plant growth but also leads to considerable yield reductions, with estimates suggesting that salinity could account for up to 50–60% of yield losses in various crops [3]. The soil salinity is further exacerbated by climate change and droughts around the globe, including Pakistan, where increased soil evaporation rates, altered precipitation patterns, and rising sea levels have been observed, leading to secondary salinization [4]. The physiological and molecular responses of plants to salinity stress, including ion imbalance, osmotic stress, and oxidative damage, require in-depth exploration to enhance crop resilience in saline environments [5].

The examination of ionomic and metabolic profiles in crops is a critical area of research, offering profound insights into the physiological and biochemical mechanisms underlying plant growth and stress responses [6]. This involves a comprehensive understanding of essential minerals and beneficial trace elements in plants, collectively called ‘ionome’ [7,8], and plays a significant role in plant growth and productivity [9,10]. It mainly involves critical ions such as Na^+^, Mg^2+^, PO_4_^3−^, K^+^, Zn^2+^, and H_4_SiO_4_, which are known to govern plant health and its response to environmental stressors, particularly salinity [11,12]. Specific metabolites, including malic acid, citric acid, glucose, fructose, and sucrose, play key roles as signaling molecules, osmolytes, or antioxidants under saline conditions [13,14,15]. Similarly, malic and citric acid aid in energy provision for essential plant processes [16,17], and glucose, fructose, and sucrose play an indispensable role in modulating osmotic balance and signal transduction [18,19,20].

Within this complex interaction of ions and metabolites, zinc (Zn) and silicon (Si) have gained special attention. Zn and Si significantly modify plant metabolite levels and ionomic balance, particularly under saline conditions [21,22]. For example, Zn is crucial for numerous physiological processes, such as enzyme activation, protein structure, and gene expression [23,24]. On the other hand, Si, recognized as beneficial, enhances plant resilience under stress, including salinity and drought by altering physiological processes (such as increased antioxidant activity and improved photosynthesis) and inducing defense responses [25,26].

The application of conventional and nanoscale fertilizers, particularly those containing Zn and Si, distinctly influences plants’ ionomic and metabolic profiles. Specifically, these fertilizers impact the concentration and distribution of key ions such as potassium, calcium, magnesium, and sulfate in plants [27,28]. Additionally, they affect the accumulation of metabolites, including sugars, OAs, and amino acids, which are crucial for plant metabolism and stress response [29]. Conventional fertilizers, though efficient, may cause an initial surge in certain ions, which may not be sustained in the long run due to leaching, volatilization, runoff, and degradation, along with other environmental losses [30,31]. In contrast, nano-fertilizers, with particle sizes within the 1–100 nm range, lead to a more stable and balanced ionomic profile in plants due to their slow nutrient release, particularly under salinity stress conditions [32,33]. This enhanced stability supports plant metabolic processes over a longer period, positively affecting the accumulation and regulation of key metabolites such as malic acid, citric acid, glucose, fructose, and sucrose, which are crucial for plant growth and resilience in response to salinity stress [34].

Therefore, this study explores the effects of conventional as well as nano-Zn and nano-Si fertilizers on the ionomic and metabolic profiles of maize under saline and non-saline conditions. The primary aim is to investigate the specific role of nano-Zn and nano-Si in enhancing maize acclimation to salinity stress at the subcellular level. It is proposed that nano-Zn and nano-Si will modulate subcellular ionomic and metabolite dynamics, thereby improving maize’s resistance to salinity by facilitating effective nutrient penetration and interaction with cellular processes.

## 2. Results

### 2.1. Effects of Nano- and Conventional Treatments on Ionic Parameters in Maize

#### 2.1.1. Na^+^ in Root and Shoot of Maize

Nano-Zn reduced Na^+^ by 35%, while conventional Zn resulted in a 31% reduction, compared to decreases of 34% and 24% with the application of nano- and conventional Si, respectively, while under saline conditions, this reduction in Na^+^ was more pronounced (Figure 1a). There was also an overall decline in root Na^+^ with Zn and Si application; nano-Zn led to a 32% decrease, whereas conventional Zn exhibited a 27% reduction. Nano-Si accounted for a 30% decrease, with conventional Si close behind, at a 21% decrease. The effect of saline treatment on root sodium was more severe while use of nano-Zn, led to significant reductions in Na^+^. Similarly, the impact of nano-Si was evidently pronounced, and a notable reduction was observed in Na^+^ in the root of maize plants under salt stress (Figure 1b).

#### 2.1.2. K in Shoot and Root of Maize

The plants not exposed to salts, significantly enhanced K^+^ content with Zn and Si treatments, compared to control plants; however, nano-treatments of both Si and Zn resulted in a substantial increase in K^+^ in the shoot (Figure 1c). Under saline conditions, nano-Zn again resulted in the highest increase of 176% in potassium content, while nano-Si application resulted in an increase that was almost double that of salt-stressed plants, and this increase was significantly higher when compared with conventional forms of both Zn and Si applications. Root analysis revealed that nano-Zn under non-saline conditions increased potassium content by 128%, whereas conventional Zn exhibited a 94% increase (Figure 1d). Nano-Si also led to an increase of 162%, whereas conventional Si showed a 47% increase. While, under saline conditions, the improvement in K^+^ content was more pronounced and nano-Zn treatment showed a 158% increase.

#### 2.1.3. K/Na in Shoot of Maize

Under saline conditions, Zn application improved the K/Na ratio in the shoot; however, nano-Zn treatment resulted in a threefold increase in the K/Na ratio compared to their saline control (Figure 1e). Nano-Si treatments were remarkably effective, tripling the control’s K/Na ratio under non-saline conditions and achieving a fivefold increase under saline conditions. Conventional Si, while yielding the least enhancement among the treatments, still managed to surpass the control by approximately 19% under non-saline conditions. In the root, under saline conditions, nano-Zn led to a more than twofold increase in the K/Na ratio; similarly, nano-Si showed the highest increment in the K/Na ratio. On the other hand, the results of conventional treatments were less pronounced (Figure 1f).

#### 2.1.4. Zn and Si Concentrations in Maize Shoot

The nano-Zn treatment under saline conditions showed an increase of almost threefold compared to without nano-Zn, while conventional Zn almost doubled the Zn concentration in the shoot (Figure 2a). On the other hand, the Si content with nano-Si was more than double whereas conventional Si showed a significant increase in Si content, with a value of approximately 134% higher than the control (Figure 2b).

### 2.2. Apoplast and Symplast Concentrations of Anions

#### 2.2.1. Nitrate (NO_3_^−^)

In apoplastic fluid, nano-Zn resulted in a 73% increase in NO_3_^−^ concentrations in non-saline and a 107% rise in saline environments when compared to controls. Conventional Zn enhanced NO_3_^−^ levels in both environments. Nano-Si significantly boosted NO_3_ concentrations by 110% in non-saline and 121% in saline environments, mirroring the positive effects observed with conventional Si (Figure 2a). For symplastic nitrate levels, nano-Zn and conventional Zn led to increases of 93% and 73% under saline conditions, respectively, with similar trends observed under non-saline conditions. Nano-Si and conventional Si increased NO_3_ concentrations significantly; however, there was a notable decrease of 25% and 15% under non-saline conditions. In saline environments, nano-Si led to an 85% increase in NO_3_ levels; in contrast, conventional Si resulted in a 25% increase (Figure 2b).

#### 2.2.2. Sulfate (SO_4_^2−^)

In the apoplast, control plants had sulfate concentrations of 2 mg/L under non-saline conditions and 1.5 mg/L under saline stress. Nano-Zn led to a remarkable 93% increase under non-saline conditions and a 49% increase under saline stress. Similarly, conventional Zn followed this positive effect in both conditions. Nano-Si also enhanced sulfate concentrations, with a 46% increase in non-saline and 31% in saline environments (Figure 2c). In the symplastic fluid, control plants showed sulfate levels of 2.6 mg/L in non-saline and 1.4 mg/L in saline environments. Nano-Zn notably increased these levels by 180% under non-saline conditions, with a similar positive impact under stress. Conventional Zn significantly enhanced sulfate levels in both environments. Nano-Si and conventional Si also resulted in notable improvements in sulfate levels (Figure 2d).

#### 2.2.3. Phosphate (PO_4_^3−^)

The control plants had phosphate concentrations of 10 mg/L and 8 mg/L in non-saline and saline environments (Figure 2f). Nano-Zn increased phosphate by 53% under non-saline and 87% under saline conditions. Conventional Zn resulted in a 25% increase under non-saline conditions and a 47% increase under saline conditions. Nano-Si and conventional Si led to increases of 27% and 24% under non-saline conditions, respectively, with both showing a similar trend of increased effects under saline conditions (Figure 2e). In symplast fluid, nano-Zn and conventional Zn showed more modest increases, with nano-Si and conventional Si also demonstrating improvements, especially notable under saline conditions for nano-Si. Conventional Si resulted in a slight increase in non-saline and an 8.77% increase in saline environments.

#### 2.2.4. Chloride (Cl^−^)

The Cl^−^ concentration in the apoplast changed significantly with the application of Si and Zn but in the same fashion (Figure 2h). The Cl^−^ concentration of 45 mg/L under non-saline conditions and 91 mg/L under saline conditions was observed, which decreased to 35 and 68 mg/L under non-saline and saline conditions, respectively, with the application of nano-Zn, compared to 41 and 72 mgL^−1^ with the conventional Zn fertilizer (Figure 2g). In symplast, the Cl^−^ level was higher than the apoplast and went above 221 mg/L under saline conditions. The nano-fertilizer of Zn and Si played a beneficial role in reducing the Cl^−^ concentration by 21% and 29% under saline conditions, compared to the control. Conventional sources of Zn and Si reduced Cl^−^ ion concentration by 7% and 21%.

### 2.3. Apoplast and Symplast Concentrations of Cations

#### 2.3.1. Sodium (Na^+^) and Potassium (K^+^)

In both the apoplast and symplast under saline conditions, nano-Zn showed a significant decrease in Na^+^, nearly 1.6 times lower than control. Similarly, nano-Si exhibited a significant decrease, around 1.7 times more than the control. For conventional treatments (conventional Zn and Si), the pattern of decrease was similar in both saline and non-saline conditions (Figure 3a,b). Nano-Si showed a notable increase in potassium uptake under non-saline conditions, approximately two times higher than the control. Conversely, under saline conditions, nano-Zn displayed the most significant increase, roughly 1.2 times higher than the control. The same pattern was observed when using conventional treatments but was less pronounced (Figure 3c,d).

#### 2.3.2. Zinc (Zn^2+^) and Silicon (H_4_SiO_4_)

Under non-saline conditions, nano-Zn enhanced zinc content to almost three times the control level, whereas conventional Zn increased it by approximately 1.4 times. In saline environments, nano-Zn sustained the zinc content at almost 2.75 times the control, while conventional Zn treatments increased it to roughly 1.6 times the control. Despite the saline challenge, nano-Zn notably surpassed conventional Zn in enhancing zinc accumulation in plant (Figure 3e,f). In both the apoplast and symplast regions, silicon content under saline conditions increased almost more than doubled, while higher Si was observed in apoplast compared to symplast. Similarly conventional Si application also resulted substantial increase in Si but relatively less pronounced than Si nonofertilizer (Figure 3g,h).

### 2.4. Determination of Sugar Contents in Apoplast and Symplast

#### 2.4.1. Glucose (C_6_H_12_O_6_)

Under normal conditions, glucose concentrations in the apoplast were markedly elevated by nano-Zn, more than sevenfold compared to the control, followed by conventional Zn and nano-Si, which prompted a sixfold rise. Under saline conditions, nano-Zn resulted in an increase of 83%, while nano-Si exhibited a 93% increase in glucose levels in the apoplast. Meanwhile, conventional Zn and Si resulted in increases of about 62% and 78%, respectively (Figure 4a). In the symplast, nano-Zn significantly increased the glucose concentration by around 90%. For saline conditions, Zn showed a substantial change in glucose levels, with an increase of 126% and 137% with nano- and conventional Zn, respectively. Similarly, Si also led to an increase of roughly 57% under saline stress (Figure 4b).

#### 2.4.2. Fructose (C_6_H_12_O_6_)

Under non-saline conditions, nano-Zn resulted in a more than fivefold increase in fructose concentration in the maize apoplast. Conventional Zn also led to a significant increase, more than tripling the concentration. Nano-Si nearly tripled the fructose level and it was doubled with conventional Si application. For saline conditions, the increases were less pronounced; however, nano-Zn still achieved a significant increase of 44% (Figure 4c). Under non-saline conditions, application of Si and Zn as nano fertilizers significantly increased the fructose concentration in the symplast, by around twofold. Under saline conditions, both nano-Zn and nano-Si resulted in considerable increases of 87% and 137%, respectively. Conventional Si showed a significant enhancement as well, with a 50% increase under saline conditions (Figure 4d).

#### 2.4.3. Sucrose (C_12_H_22_O_11_)

In non-saline environments, the control’s sucrose concentration was increased by approximately 8.5 times with nano-Zn and 4.4 times with conventional Zn. Under saline conditions, nano-Zn and conventional Zn increased sucrose levels by 36% and 34%, respectively. Nano-Si led to a 6.4-fold increase under non-saline conditions and a 32% boost under saline stress, while conventional Si resulted in a nearly sixfold increase in non-saline and a 12% increase in saline environments, compared to the control (Figure 4e). In the symplast, nano-Zn nearly doubled the concentration under non-saline conditions, with a 97.44% increase and a 68.15% increase under saline conditions. Nano-Si showed a 93.49% increase under saline conditions. Conventional Zn and Si treatments also showed increases, though they were less pronounced (Figure 4f).

### 2.5. Apoplast and Symplast Concentrations in Organic Acids (OAs)

#### 2.5.1. Malic Acid (C_4_H_6_O_5_)

The malic acid concentration in both the apoplast and symplast increased when the plant was stressed. The application of nano-Zn increased malic acid to 44 mgL^−1^ (around 147% higher than the control) compared to a 50% increase with conventional Zn under normal conditions, while under stress conditions, the malic acid concentration was 82 mgL^−1^ (118% increase) with nano-Zn treatment compared to a 14% increase with conventional Zn. Silicon did not influence malic acid levels significantly under non-saline conditions; however, under saline conditions, it increased to 54 mgL^−1^ (43% increase) with nano-Si, and only a 19% increase was observed with conventional Si application (Figure 5a). The malic acid level in the symplast exhibited a considerable elevation with nano-Zn, reaching 217 mgL^−1^, compared to 134 mgL^−1^ when conventional Zn was applied under saline conditions. The Si application also showed an increase in symplastic malic acid, up to 125 mgL^−1^ with a conventional source, but it increased to 173 mgL^−1^ with a nano-fertilizer of Si under stress conditions (Figure 5b).

#### 2.5.2. Citric Acid (C_6_H_8_O_7_)

Zn application influenced the citric acid concentration in the apoplast of maize leaf under normal as well as salt stress conditions. Plants treated with nano-Zn showed higher levels (5 mgL^−1^) compared to 3 mgL^−1^ with conventional Zn application under normal conditions, and this increased to 9 mg/L under saline conditions. Si application also increased the levels but not significantly, especially under stress conditions (Figure 5c). The symplastic citric acid concentration also increased under stress conditions, but nano-Zn application in plants showed a more notable increase of 15 mgL^−1^. Nano-Si showed almost the same trend but caused higher levels of citric acid under normal conditions (37% over control) Figure 5d.

#### 2.5.3. Oxalate (C_2_O_4_^2−^)

In apoplastic fluid, nano-Zn treatment slightly increased oxalate under non-saline conditions but decreased it by 23% under saline stress. Conventional Zn showed a 14% increase in non-saline and a 12% decrease under saline conditions. Nano-Si led to decreases of 19% and 11% under non-saline and saline conditions, respectively. Conventional Si significantly increased oxalate by 59% under non-saline conditions (Figure 5e). In symplastic fluid, nano-Zn increased oxalate by 38.02% in non-saline environments but decreased it by 38% under saline stress. Conventional Zn saw a 94% rise in non-saline environments but saw a notable decrease under saline stress. Nano-Si led to a 52% increase under non-saline conditions and a 33% decrease in oxalate in saline environments. Similarly, conventional Si raised oxalate by 61% in non-saline environments compared to a 28% reduction under saline conditions (Figure 5f).

### 2.6. Heatmap Correlation Graph for Different Parameters

#### 2.6.1. Heatmap for Ion Distribution in Whole Plants

A heatmap was utilized to display the concentrations of ions, providing a clear visual differentiation between treatments and conditions (Figure 6). A color gradient from blue to red indicates lower to higher ion concentrations, allowing for an immediate visual comparison of how each treatment influences ion uptake and distribution. The map clearly indicates that nano-fertilizer treatments enhance ion accumulation more effectively than conventional fertilizers, a pattern consistent in both minerals and stress conditions. The thresholds for color transitions from blue to red were set relative to the minimum and maximum recorded values for each ion, which provides a standardized scale across different ions for easy comparison. A pronounced increase was observed in the K^+^ concentration in maize treated with nano-fertilizers, as depicted by the shift from blue to deep red, compared to the conventional treatments. This suggests a more efficient uptake mechanism for K^+^ prompted by nano-fertilizers, which could be instrumental in enhancing the plant’s stress tolerance but still needs further evidence to understand the underlying mechanism.

#### 2.6.2. Sugar Contents, Organic Acids, and Anions in the Apoplast and Symplast

These heatmaps contrast the ion concentrations within the apoplastic and symplastic compartments of plants, underlining the differential uptake and transport mechanisms influenced by nano- and conventional fertilizer treatments in saline and non-saline environments (Figure 7). The color gradients, transitioning from blue to red, quantify ion levels from low to high. In the apoplast heatmap, citric acid (C.A) shows a significant response to nano-fertilizer, with a stark transition from blue to red, highlighting a major uptake. In the symplast heatmap, malic acid (M.A) displays a prominent increase with nano-fertilizer treatment, indicated by a clear shift to red. Nano-fertilizers demonstrated a consistent superiority in ion enrichment across both compartments, suggesting more efficient ion assimilation and internal distribution.

### 2.7. Scatterplot Matrix (SPLOM) Analysis

The SPLOM facilitated the examination of mineral concentrations in maize under varying environments. Under both saline and non-saline conditions, Na^+^ concentrations showed a strong positive correlation between shoots and roots, suggesting a consistent Na^+^ distribution regardless of external salinity (Figure 8). However, the K^+^ levels in shoots were inversely related to Na^+^ levels in roots (Corr. −0.777), with this relationship being more pronounced under saline conditions, hinting at the plants’ adaptive mechanisms to salinity stress. The K/Na ratios in roots and shoots were strongly and positively correlated with root K^+^ levels, with correlation coefficients of 0.996 and 0.963, respectively. These ratios were notably higher in the nano-Zn and nano-Si treatments, indicating a potential synergistic effect of nanoparticle application on ion homeostasis. In contrast, the correlation between the shoot’s zinc concentration and the K/Na ratios in roots was weaker (Corr. 0.534), with the least variation observed in the conventional Si application, suggesting a more complex interaction possibly modulated by the form of silicon applied. The density plots revealed variable distribution patterns for each mineral, with the root Na^+^ concentration displaying a skewed distribution, especially under saline conditions, potentially reflecting a stress-induced accumulation pattern. These data imply that the NPs of Zn and Si seem to influence the ionic balance favorably, which may have implications for enhancing plant resilience to salinity stress.

## 3. Discussion

### 3.1. Zinc and Silicon Nano-Fertilizers under Salt Stress

The application of zinc and silicon nano-fertilizers helped maize to withstand salt stress, as evidenced by the significant improvements in the ionomic and metabolite profiles. Zn plays a crucial role in enzymatic activity for protein synthesis and stress resistance [35], as well as enhancing K discrimination over Na in plant tissues [36,37]. This modulation assists in maintaining ionic balance and easing Na toxicity, thereby supporting plant growth under saline conditions [38]. Moreover, Zn and Si application also reduces oxidative stress by influencing osmoregulation and antioxidant activity [39]. Si has also been instrumental in reinforcing the structural integrity of cell walls [40,41], thus restraining Na in the apoplast and symplast [42]. This mechanism, in turn, aids in preserving cellular homeostasis and enhancing the plant’s ability to regulate osmotic stress [43]. The nano-formulation of these fertilizers appears to enhance their bioavailability and efficient delivery compared to their conventional sources.

### 3.2. Impact on Ionomic Profile

This study showed that nano-fertilizers altered K/Na under saline conditions by increasing the potassium content up to 176% in shoots when nano-Zn was applied (Figure 1) and helped plant in regulating ionic homeostasis, osmotic adjustment to uphold its physiological processes, and ultimately yield [34,44]. Applying nano-Zn and nano-Si treatments to maize under saline conditions significantly reduced Na^+^ uptake (Figure 1a,b), likely due to nanoparticles, which improved the membrane permeability of roots [45] by interacting with lipid and protein components, affecting its fluidity and aquaporin channels and resulting in decreased Na^+^ uptake [46,47,48]. Furthermore, the significant increases in Zn and Si concentrations in maize shoots were nearly threefold for Zn and more than twofold for Si under saline conditions (Figure 2a,b), depicting that nutrients application as nanoparticles facilitates nutrient bioavailability and absorption [49]. This helps in triggering plant antioxidant defense and osmoprotectant production [50,51]. This leads to improved plant resilience to salt stress by enhancing osmotic regulation and facilitating cellular detoxification [52]. The distinct ionic profiles observed in the apoplast and symplast heatmaps reflect the physiological responses of plants to fertilizer treatments, indicating that nano-fertilizers may alter ion compartmentalization, potentially enhancing stress resistance (Figure 6 and Figure 7).

### 3.3. Effect on Metabolite Profile

#### 3.3.1. Ion Dynamics and Plant Stress Response

Under salinity stress, the role of anions like phosphate, sulfate, nitrate, and chloride becomes crucial for plant survival and adaptation [53], enabling plants to maintain vital processes under stress [54]. Sulfate plays a key role in synthesizing glutathione, an antioxidant that helps protect against oxidative damage induced by salinity [55]. Nitrate, as a primary nitrogen source, supports osmoregulation and the synthesis of essential biomolecules, aiding in stress adaptation [56]. Conversely, higher chloride uptake under saline conditions can be detrimental, leading to ionic imbalance and toxicity [57]. Thus, the ability of plants to regulate the uptake and compartmentalization of these anions is critical for mitigating salinity stress.

Both nano-Zn and nano-Si boosted NO_3_^−^ in the apoplastic and symplastic fluid under saline conditions, and this could be associated with the role of Zn in enhancing the efficiency of nitrate reductase, an enzyme crucial for the reduction of NO_3_^−^ to NH_4_^+^, which is more readily assimilated by plants by acting as a catalyst [58]. This enzymatic boost not only improves nitrate utilization in the plant but also supports better growth and stress mitigation by improving protein synthesis [59] and more efficient transport within the plant [60]. The significant elevation in nitrate levels with nano-Si, particularly under saline conditions, could be linked to its ability to mitigate oxidative stress and enhance plant vigor, thereby indirectly supporting more effective nitrate absorption and utilization [61].

Sulfate responded better to Zn and Si under non-saline conditions compared to saline conditions, which led to a more efficient conversion of inorganic sulfate to sulfide. Similarly, nano-Zn significantly increased phosphate levels, by 87%, under saline conditions (Figure 3e), by reducing the fixation of phosphate and making it more accessible to the plants [62]. Plants treated with nano-Zn may exhibit increased root exudation of OAs and other compounds that mobilize phosphate by chelating soil minerals and releasing bound phosphate [63]. This enhanced exudation can improve phosphate acquisition, especially in saline soils, where phosphate availability is often reduced due to precipitation with cations like calcium and magnesium [64]. On the other hand, nano-Zn promotes the selective uptake of essential nutrients over Cl^−^ ions due to competitive interaction at the root membrane level [65]. Zinc presence may influence membrane transporters to favor the uptake of Zn^2+^ ions and other beneficial ions, reducing the passive influx of Cl^−^ ions into the plant system [66,67]. While nano-Si can form a silica layer around the roots, acting as a physical barrier that reduces the uptake of toxic ions, including Cl^−^ [68].

#### 3.3.2. Sugar Metabolism and Osmotic Balance

Plants accumulate sugars and other organic compounds to serve as osmolytes to combat osmotic stress [69]. These sugars, especially glucose and fructose, are crucial for providing energy and resilience under adverse conditions [70]. In our study, nano-Zn and nano-Si treatments under saline conditions significantly enhanced glucose, fructose, and sucrose levels in maize (Figure 4). Specifically, glucose in the apoplast increased by 83% with nano-Zn and 93% with nano-Si. This sugar accumulation, facilitated by improved nutrient uptake and photosynthetic efficiency due to the application of Zn and Si as nanoparticles, supported osmotic regulation and stress resistance [71]. The adjustment in ionic balance as lower sodium levels and higher potassium levels, enhancing the K/Na ratio, contributed to the observed increase in sugar content, illustrating the critical role of nano-fertilizers in promoting mitigation to salinity.

OAs, such as malic, citric, and oxalic acids, are integral to the tricarboxylic acid (TCA) cycle and also play critical roles for plants, including osmoregulation, energy production, and stress signaling [13,72]. Oxalic acid, through its chelating properties, can influence the plant’s ion balance and stress responses [73]. Salinity stress disrupts plant water and ion homeostasis, prompting adjustments in organic acid metabolism [74]; thus, the biosynthesis, secretion, or distribution of OAs are affected in response to stress conditions, and multiple genes are either upregulated or downregulated to fine-tune the adaptation to these stressed conditions as part of the plant’s strategy to mitigate stress effects, including oxidative damage but at the cost of energy [75,76]. This crosstalk between plant metabolites and abiotic stresses has positive impacts on plant growth and enhances its tolerance and immunity, as confirmed in this study during the metabolomic profiling of sugars and OAs accumulated under stress conditions. The data show that, under saline conditions, the sucrose and oxalate concentrations in the apoplast increased, and this was further elevated with fertilizer treatment (Figure 4 and Figure 5). Nanoparticles significantly increased citric acid and oxalate concentrations under saline conditions. This increase can be explained by the activation of stress-responsive signaling pathways [77], which play roles in stress adaptation by regulating the synthesis and accumulation of citric acid and glycolate oxidase (GOX), which are involved in the biosynthesis and metabolism of oxalic acid [78]. Nano-Zn facilitates the activation of antioxidant defense pathways, such as those mediated by glutathione S-transferases (GSTs) and superoxide dismutases (SODs), which contribute to the detoxification processes and stress mitigation [79]. Silicon is not directly involved in metabolic reactions but improves plant resilience by strengthening cell walls and supporting osmotic adjustment and stress signaling pathways [80].

Zn and Si applications help plants, especially under stress conditions, but optimal levels are crucial to consider for better and economical plant growth and productivity. The proper ionic balance facilitated by an optimal K/Na ratio can contribute to improved physiological processes in the maize plant, leading to better overall yield and quality under various growing conditions, especially in saline environments. However, exceeding the optimal concentrations of ions and OAs, which include essential nutrients such as phosphate and sulfate, can disrupt maize metabolic processes and ion balance, as well as hinder cellular respiration and cause toxicity to the plant. The enhanced effectiveness of nano-fertilizers can be attributed to their unique properties, such as smaller particle size, leading to better soil penetration, increased surface area for more efficient interaction with plant roots, and improved uptake and utilization of nutrients [81]. These factors contribute to a more effective delivery of essential nutrients, facilitating significant improvements in plant physiological parameters under saline conditions. The findings from our study underscore the significant potential of zinc and silicon nano-fertilizers in transforming agricultural practices, particularly for crops grown in salt-affected soils. These nano-fertilizers offer a promising strategy for sustaining agriculture in challenging environments, enabling farmers to achieve higher yields and more resilient crops [81]. The one question that arises in this study and remains unanswered is what is the optimal level of these sugars and OAs in maize, and how their levels influence the product (maize) quality? So, it is suggested that future studies in this area may be helpful to understand and answer such questions.

## 4. Materials and Methods

### 4.1. Synthesis and Characterization of Nanoparticles

The ZnO nanoparticles were developed through a precipitation approach, using a mixture of 0.1 M NaOH and 0.05 M ZnSO_4_∙7H_2_O, and the reaction was sustained at 80 °C for 2 h [82]. The SiO_2_ nanoparticles were synthesized via a sol–gel technique, maintaining a TEOS–ethanol ratio of 1:4 at room temperature for 24 h [83]. The ZnO nanoparticles underwent a calcination step after post-synthesis, while the SiO_2_ nanoparticles were subjected to aging, drying, and calcination processes. Comprehensive characterizations of these nanoparticles were conducted at the Department of Materials Science, Kiel University, Kiel, Germany. Techniques including ultraviolet–visible spectroscopy (UV-6000, R&M, Cambridge, UK), X-ray diffraction (Jupiter, Oxford Instruments, Abingdon, UK), and scanning electron microscopy (Supra 55VP, Zeiss, Oberkochen, Germany) were utilized as described in [84]. For transmission electron microscopy (TEM) analysis, the samples were finely ground, dispersed in n-Butanol, and placed on Cu lacey TEM grids (Tecnai G2 F30 S Twin 300 kV/FEG, FEI Company, Eindhoven, The Netherlands), equipped with an EDX detector. The XRD results confirmed the crystalline structure of the nanoparticles, with average sizes measured at approximately 15 nm for SiO_2_ and 12 nm for ZnO nanoparticles. SEM and TEM imaging revealed particle sizes ranging between 10 and 20 nm, aligning with the nanoscale dimensions of the particles, which is described in detail in a previous study [84].

### 4.2. Experimental Setup

A hydroponic experiment was conducted in growth rooms having a 14 h photoperiod with light intensity ≥ 350 μmol m^−2^ s^−1^ PAR, day/night temperatures of 27/16 °C, and 70% relative humidity. Maize seeds were soaked overnight in a 0.5 mM CaSO_4_ solution in plastic trays, and light was provided by a combination of red and blue LED lights for full-spectrum illumination. Seedlings, at the two-leaf stage, were transplanted into 5 L pots containing a modified Hoagland’s nutrient solution, adjusted to a quarter strength to reflect recent research findings and best practices in plant nutrition. Treatments included a control (only the solution), nano-treatments (ZnO and SiO_2_) and conventional treatments (ZnSO_4_ and K_2_SiO_3_) of Zn and Si @ 10 ppm and 90 ppm, respectively. Half of the plants were also subjected to salinity stress using 100 mM NaCl, added gradually with 25 mM daily increments. All the experimental units were replicated three times. Nutrient solution pH was maintained at 6.0 ± 0.5 and refreshed biweekly with continuous artificial aeration.

### 4.3. Measurements

#### 4.3.1. Ion Extraction from the Leaf Apoplast and Symplast of Maize Plant

The infiltration–centrifugation method was used to extract apoplastic fluid [85]. A razor blade was used to trim the leaves, and deionized water was used to properly wash them. When undamaged leaves were penetrated, the pressure was decreased to an estimated 20 kPa by pressing the plunger on plastic syringes (60 mL) loaded with 40 mL of deionized water. The leaves were then centrifuged at 327 g for five minutes at 4 °C. The collected fluid is referred to as the apoplastic washing fluid (AWF). AWF samples were kept in storage at −80 °C for analysis. The ion concentration in the apoplastic of field maize leaves was calculated by multiplying it with the dilution factor. Following the isolation of AWF, the remaining leaf tissue was centrifuged at 327 g for 10 min after being shock-frozen in liquid nitrogen at 70 °C. The leaf sap was then kept at −80 °C until analysis and referred to as the symplastic leaf fraction [85].

#### 4.3.2. Determination of Ion Concentration Determination in Apoplastic and Symplastic Leaf Fractions

Chloroform was added to the samples, and the clear supernatant was collected after centrifugation. Ion chromatography (Dionex, DX2500, Idstein, Germany) on an IonPac anion exchange column (Dionex, AS9) was used to determine the ion concentration and sugar content such as chloride ion, nitrate, malic acid, sulfate, oxalate, phosphate, and citric acid in the leaf symplast and AWF [86].

#### 4.3.3. Ionic Determination in Whole-Leaf Samples

Dry plant samples of 200 mg digested with 10 mL HNO_3_ in a microwave oven. After digestion, samples were cooled down and diluted up to 100 mL, and micronutrients (Zn and Si) and macronutrients (Na and K) were measured by inductively coupled plasma mass spectroscopy (ICP-MS; Agilent 7700, Santa Clara, CA, USA) as used in [87].

#### 4.3.4. Si Determination in Leaf Samples

For Si determination, an oven-induced digestion (OID) method was utilized [88]. Briefly, 100 mg of dry and ground leaf samples were added with five drops of octyl alcohol and 2 mL of 30% H_2_O_2_ and subsequently heated at 95 °C in a convection oven. After 30 min, 4 mL of 50% NaOH was introduced to the hot samples and vortexed and then returned to the oven at the same temperature. This digestion process continued for 4 h to ensure a thorough breakdown of organic matter [89]. After 4 h, 1 mL of 5 mM NH_4_F was added to facilitate the formation of monosilicic acid. The prepared sample was subjected to Si determination using the molybdenum blue method [89], and absorbance was measured at 650 nm using ultraviolet–visible spectroscopy (UV-6000, R&M, Cambridge, UK).

### 4.4. Statistical Analysis

Statistical analyses were performed using Statistix 10.1 software, adhering to the methodology outlined by the authors of [90]. Post hoc comparisons were conducted via the least significant difference (LSD) test at a 5% significance level to identify meaningful differences between treatment means. Data visualization was executed using MS Excel, where the standard error of the mean is depicted as error bars in the graphical representations.

## 5. Conclusions

Conclusively, our investigation into the effects of zinc and silicon nano-fertilizers on maize under salinity stress reveals a promising avenue for enhancing crop resilience in saline environments. By significantly modulating the ionomic and metabolite profiles, these nano-fertilizers demonstrated a superior capability over conventional fertilizers in improving plant growth, nutrient uptake, and stress resistance. The distinct improvements in key physiological parameters, such as the K/Na ratio, phosphate, sulfate, and nitrate levels, alongside the reduction in detrimental Cl^−^ ions and the augmentation of crucial OAs and sugars, underscore the multifaceted benefits of nano-fertilizer application. These findings not only highlight the potential of nano-Zn and nano-Si to mitigate the adverse effects of salinity on crops but also suggest a transformative impact on agricultural practices for salt-affected soils. Future research that delves deeper into the molecular mechanisms and gene signaling pathways influenced by nano-fertilizers will be pivotal in fully harnessing their potential for sustainable agriculture.

## Figures and Tables

**Figure 1 plants-13-01224-f001:**
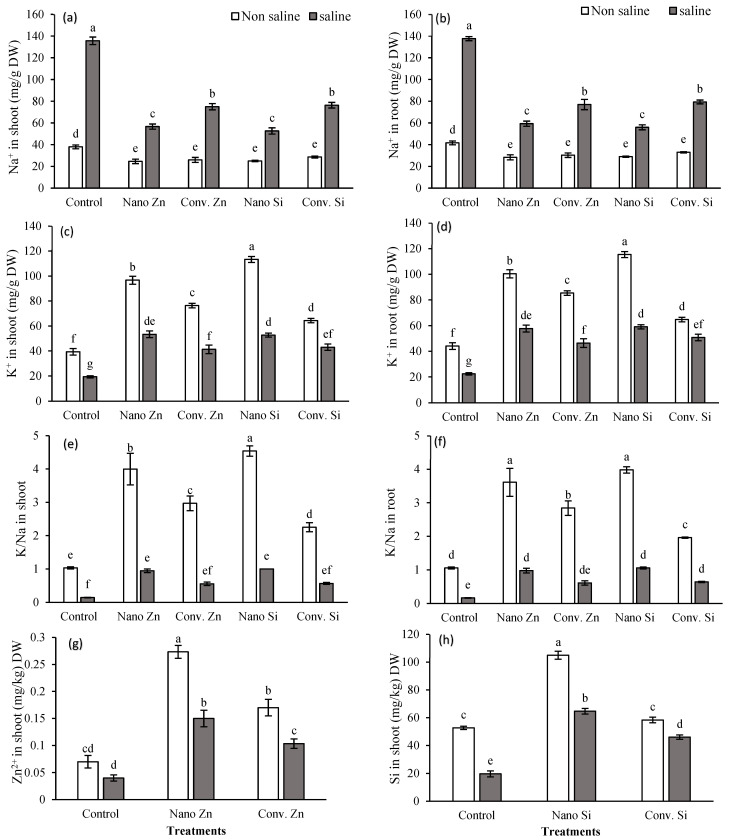
Sodium in shoot and root (**a**,**b**), potassium in soot and root (**c**,**d**), K/Na ration in shoot and root (**e**,**f**) and zinc in shoot and root (**g**,**h**) of maize plant under saline and non-saline soil (100 mM NaCl) along with exogenous application of Zn@100 ppm and Si@90 ppm both as conventional fertilizers (ZnSO_4_ and K_2_SiO_3_) and nanofertilizers (ZnO and SiO_2_). Means with distinct letters differ significantly (LSD test; *p* < 0.05).

**Figure 2 plants-13-01224-f002:**
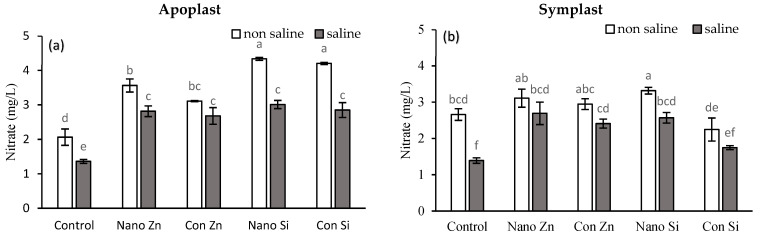
Nitrate, sulphate, phosphate, and chloride in apoplast (**a**,**c**,**e**,**g**) and symplast (**b**,**d**,**f**,**h**) of maize plant under salt (100 mM NaCl) stress with the application of Zn@100 ppm and Si@90 ppm both as conventional fertilizers (ZnSO_4_ and K_2_SiO_3_) and nano-fertilizers (ZnO, SiO_2_). Means with distinct letters differ significantly (LSD test, *p* < 0.05).

**Figure 3 plants-13-01224-f003:**
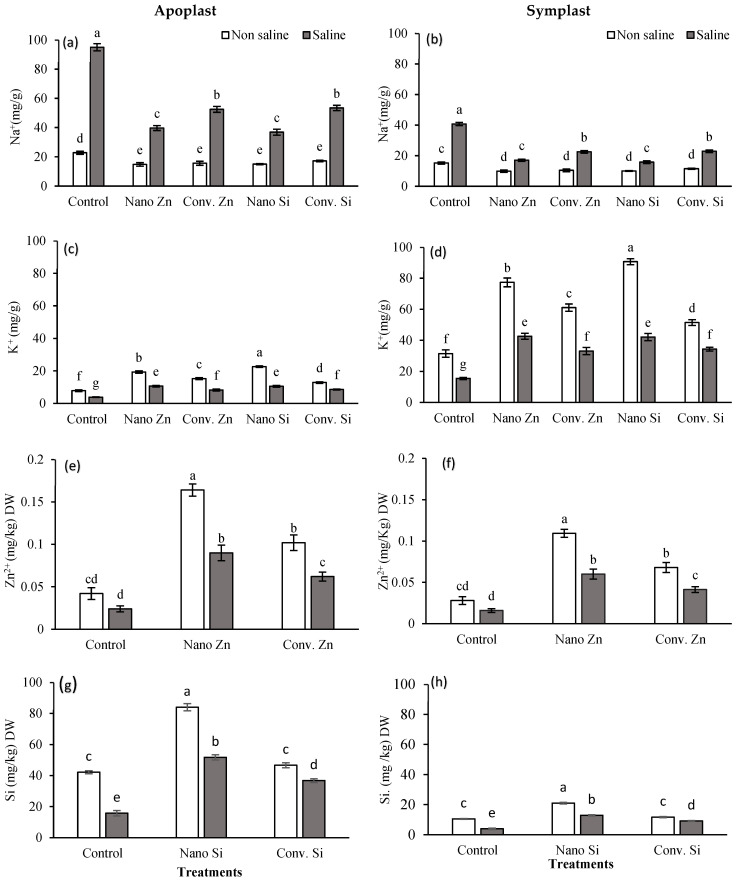
Sodium, potassium, zinc, and silicon concentrations in apoplast (**a**,**c**,**e**,**g**) and symplast (**b**,**d**,**f**,**h**) of maize plant under salt stress (100 mM NaCl) with the application of Zn@100 ppm and Si@90 ppm both as conventional fertlizers (ZnSO_4_ and K_2_SiO_3_) and nanofertlizers (ZnO, SiO_2_). Means with distinct letters differ significantly (LSD test, *p* < 0.05).

**Figure 4 plants-13-01224-f004:**
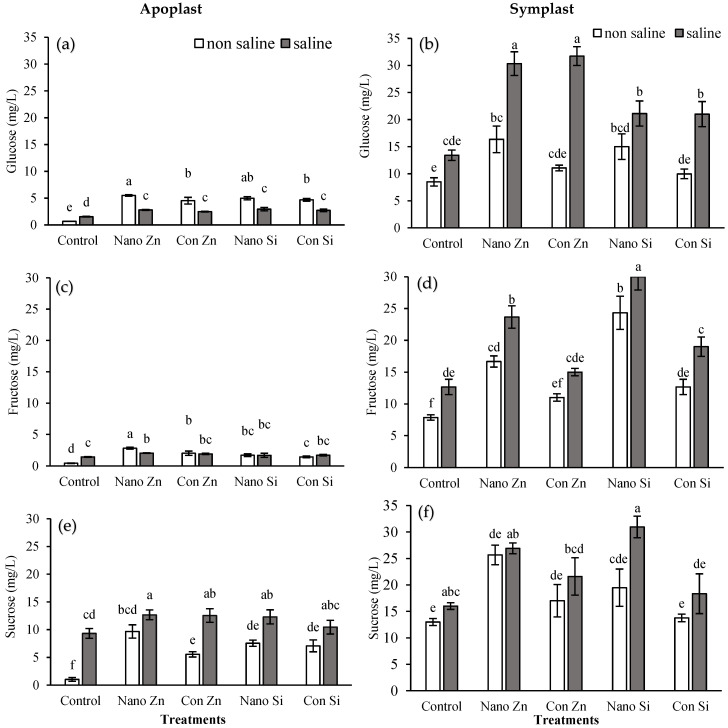
Glucose, fructose, and sucrose concentrations in apoplast (**a**,**c**,**e**) and symplast (**b**,**d**,**f**) of maize plant under saline and non-saline soil (100 mM NaCl) along with exogenous application of Zn@100 ppm and Si@90 ppm both as conventional fertilizers (ZnSO_4_ and K_2_SiO_3_) and nanofertlizes (ZnO and SiO_2_). Means with distinct letters differ significantly (ANOVA, LSD, *p* < 0.05; SE bars, n = 3).

**Figure 5 plants-13-01224-f005:**
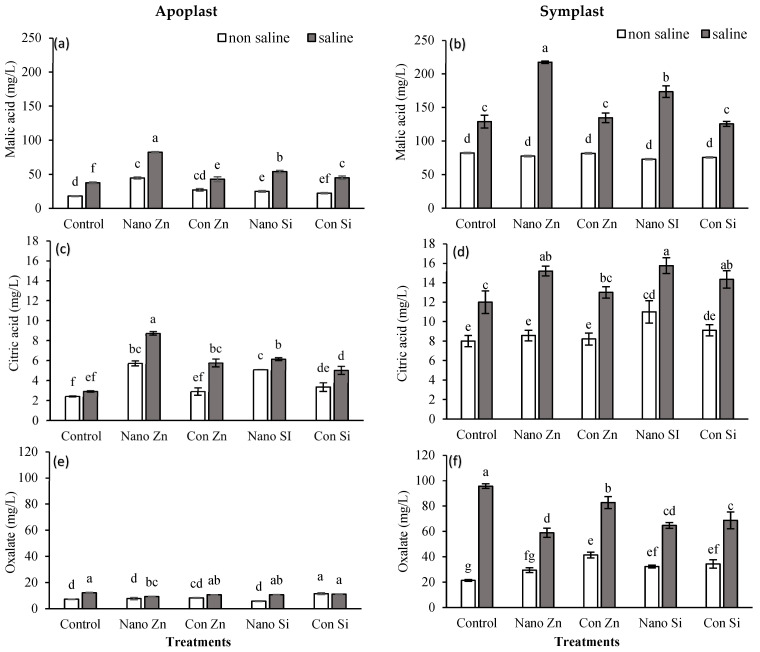
Malic acid, citric acid, and oxalates in apoplastic (**a**,**c**,**e**) and symplastic (**b**,**d**,**f**) of maize leaf fractions under normal and saline (100 mM NaCl) conditions along with the exogenous application of Zn@100 ppm and Si@90 ppm both as conventional fertlizers (ZnSO_4_ and K_2_SiO_3_) and nanofertilizers (ZnO and SiO_2_). Means with distinct letters differ significantly (ANOVA, LSD, *p* < 0.05; n = 3).

**Figure 6 plants-13-01224-f006:**
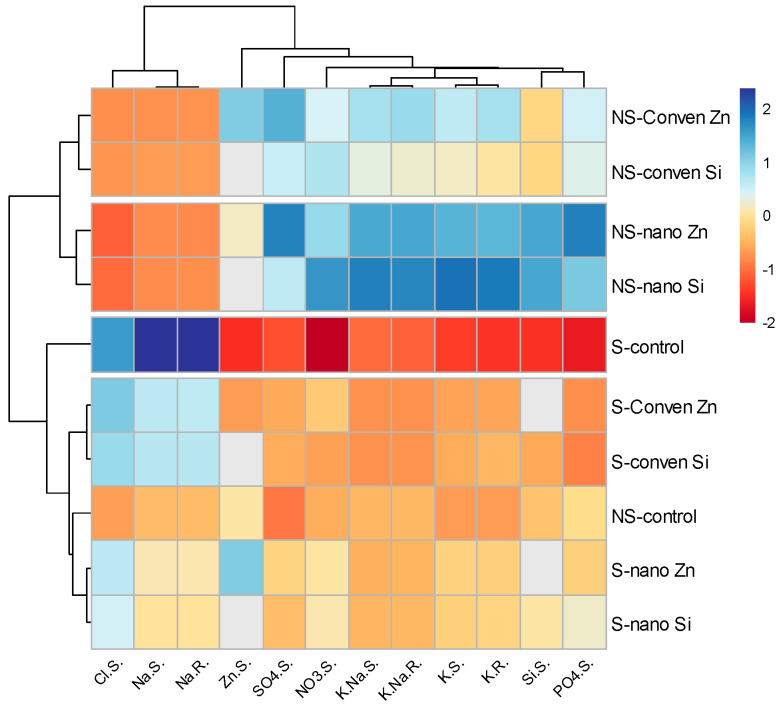
Relationship heatmap of ions in leaves. The figure includes potassium in roots (K.R.), potassium in shoots (K.S.), the potassium–sodium ratio in shoots (K.Na.S), the potassium–sodium ratio in roots (K.Na.R), silicon in shoots (Si.S), zinc in shoots (Zn.S), sodium in roots (Na.R), sodium in shoots (Na.S), sulfate in shoot (SO_4_.S), nitrate in shoot (NO_3_.S), phosphate in shoot (PO_4_.S), and chloride in shoot (Cl.S.).

**Figure 7 plants-13-01224-f007:**
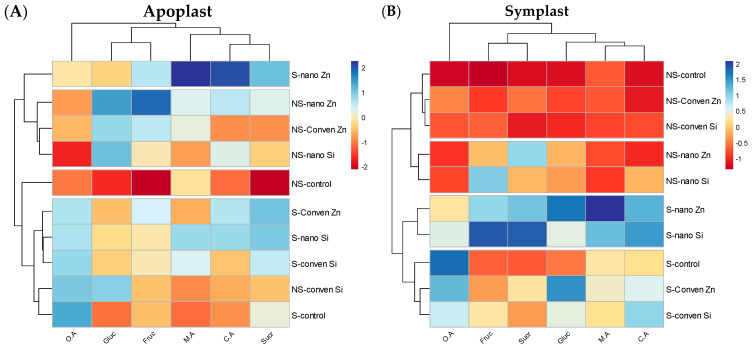
Comparative heatmaps of citric acid (C.A), malic acid (M.A), oxalic acid (O.A), glucose (Gluc), fructose (Fruc), and sucrose (Sucr) in the apoplast (**A**) and symplast (**B**) of plants treated with conventional and nanofertilizers of Zinc and Silicon.

**Figure 8 plants-13-01224-f008:**
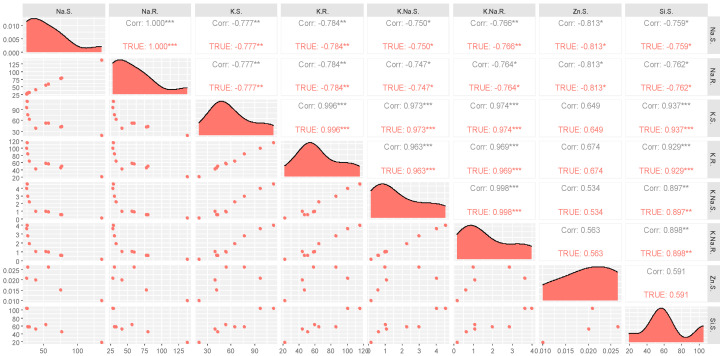
Correlation matrix of ions in plant tissues under saline and non-saline conditions. This figure includes potassium in roots (K.R.), potassium in shoots (K.S.), potassium–sodium ratio in shoots (K.Na.S), potassium–sodium ratio in roots (K.Na.R), silicon in shoots (Si.S), zinc in shoots (Zn.S), sodium in roots (Na.R), and sodium in shoots (Na.S). The annotations of *, **, *** on Correlation values represents significance at probability level of 0.1, 0.05 and 0.01 respectively.

## Data Availability

Data will be available upon request to the corresponding author.

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
