# Peer review of "Zinc and Silicon Nano-Fertilizers Influence Ionomic and Metabolite Profiles in Maize to Overcome Salt Stress"

_plants, 2024, doi:10.3390/plants13091224_

Round 1
Reviewer 1 Report
Comments and Suggestions for Authors
In this manuscript, the authors evaluate the effects of Nano Zn and Si fertilizer on maize under salt stress by quantifying various ions, sugars and organic acid metabolites of the maize under different growth conditions.
Major points:
1. The manuscript appears to be written and assembled carelessly. Starting from Figure 2, the text and figures do not match. Every figure is shifted (Figure 4 is supposed to be Figure 2; Figure 5, Figure3; and so on). In addition, in section 2.6, the explanation of the heatmap (Line 303-306) should describe sugar contents, organic acid and anions, but only ions are mentioned. The logically, the anion analysis should be combined with the cation analysis in Figure 6 and the heatmap in Figure 7 should focus on sugar contents and organic acids.
2. In the introduction (Line 57), P and Si usually exist in plants as inorganic anions. For the sentence to be correct chemically, the reviewer recommends to write all the elements in the sentence in their ionic forms, for example, Na+, Mg2+.
3. The manuscript provided data that shows nano-fertilizer facilitates ion uptake and accumulation of sugar and organic acid contents in maize. However, are these nutrients the more the better for maize? The authors did not discuss whether higher concentration of these ions and organic acids are better for maize quality or worse and what are the healthy ranges of these ions and sugars and organic acids.
4. The heatmaps in Figure 6 and 7 need more explanation. The text only says that the blue and red color indicate ion levels from low to high. The concentrations of different ions and nutrients are quite different. The authors need to explain more in detail how the level is set for each of the nutrients in the heat maps.
Comments on the Quality of English LanguageThere are multiple grammatical errors in the manuscript, especially in the Introduction.
Author Response
The reply of the Comments from Reviewer-1 is attached.

Reviewer 2 Report
Comments and Suggestions for Authors
The paper titled “Zinc and Silicon Nano Fertilizers influence Ionomic and Metabolites Profile in maize to overcome salt stress” effectively contributes to agricultural science by exploring zinc and silicon nano-fertilizers' impacts on maize under salinity stress. The paper is commendable for its depth and contribution to agricultural sciences yet it raises intriguing questions detailed in the comments section below.
Comments
>Line 63-70: Given the focus on Zinc (Zn) and Silicon (Si) in enhancing plant resilience, are there potential negative effects of using high concentrations of these elements, especially in nano form, on plant health or soil quality?
>Line 72-83: The text suggests nano fertilizers offer a more stable ionomic profile compared to conventional fertilizers. What evidence exists to support this claim, and are there studies comparing the long-term effects of nano versus conventional fertilizers on soil health?
>Line 203-210: Write in the same font as the previous one, Times New Roman.
>Line 290-306: Given the initial decision to utilize heatmaps for data visualization in the study and acknowledging that heatmaps are indeed adequate, what prompted the additional consideration for a Scatterplot Matrix (SPLOM)? The suggestion to create a SPLOM appears to be based on the understanding that it could provide a more nuanced view of the data by exploring and visualizing the relationships between multiple variables simultaneously.
>Question (Line 334-350): How do changes in the ionomic profile, specifically the increased K/Na ratio under saline conditions, affect the nutritional quality of maize?
>Question (Line 371-384): The discussion mentions increased phosphate levels under saline conditions due to nano-Zn application. How does this affect the phosphate cycle in agricultural ecosystems, especially considering the potential for eutrophication in nearby water bodies?
>Question (Line 419-428): The enhanced effectiveness of nano-fertilizers is noted, especially under saline conditions. What are the economic implications of adopting nano-fertilizer technology on a large scale, considering the cost of production, accessibility for smallholder farmers, and potential need for new application technologies?
>Hydroponic System Setup (Lines 448-461): The transition from hydroponic to soil-based environments can significantly affect plant response to fertilizers. How might the findings from this hydroponic study translate to soil environments, especially considering the varied soil chemistries and structures?
Author Response
The Reply of Comments is attached as MS Word file.

Round 2
Reviewer 1 Report
Comments and Suggestions for Authors
The revised manuscript corrected the mismatched and missing figures in the original manuscript.
There are still a few points that the authors need to address.
1. In the introduction, Si exists in plant mainly in the form of anions, SiO3 2-. However, the data for Si is included in section 2.3 as the cation, Si4+. Do the authors have any specific reason for this? If not, it seems to the reviewer that this piece of data should be re-organized into Figure 2.
2. In Line 435-459, the authors added discussion on how concentrations of mineral and organic ions can affect yield and quality of maize. The discussion is very general. The standardized or desired concentration of these ions for high yield and quality maize are not provided. Therefore, it is still unclear whether the fertilizer treatment is improving the maize quality or not. For example, in figure 4e, the sucrose concentration in apoplast is close to 10 mg/L in saline condition and it is about 8-10 fold higher than non-saline condition. Which concentration is more normal or better for the quality of yield of the maize? Is the saline condition supposed to be not healthy? Then the fertilizer treatment in general increased the sucrose concentration in both saline and non-saline conditions. Is it desired or not? Similar questions hold for Figure 5f. In saline condition, the oxalate concentration in symplast is about 5-fold higher than non-saline condition. The fertilizer treatment increased oxalate concentration for non-saline condition but decreased it for the saline condition. Without the knowledge on the proper or healthy concentration of oxalate in maize, it is difficult for the reviewers and potentially the readers to evaluate the effect of the fertilizer.
In addition, the sentence is itself needs some grammatical editing. “However, surpassing the optimal concentrations of ions and organic acids, including essential nutrients like phosphate and sulphate can disrupt maize's metabolic processes and ion balance, hindering cellular respiration and toxicity to plants”. Is it supposed to be “disrupt ion balance, hinder respiration and cause toxicity”?
3. In figure 4f, the LSD indicator for nano Si at saline condition is missing.
Comments on the Quality of English Language/
Author Response
Thanks for your constructive and valuable comments. All the comments and suggestion are replied (file attached) and also incorporated, where needed in the revised version of the manuscript.
